# Annexin A6 Polymorphism Is Associated with Pro-atherogenic Lipid Profiles and with the Downregulation of Methotrexate on Anti-Atherogenic Lipid Profiles in Psoriasis

**DOI:** 10.3390/jcm11237059

**Published:** 2022-11-29

**Authors:** Fuxin Zhang, Ling Han, Bing Wang, Qiong Huang, Nikhil Yawalkar, Zhenghua Zhang, Kexiang Yan

**Affiliations:** 1Department of Dermatology, Huashan Hospital, Fudan University, Shanghai Institute of Dermatology, Shanghai 200040, China; 2Department of Dermatology, Inselspital, Bern University Hospital, University of Bern, CH-3000 Bern, Switzerland

**Keywords:** psoriasis, Annexin A6, lipometabolism, *rs11960458*, methotrexate

## Abstract

Background: Annexin A6 (AnxA6) is a lipid-binding protein that regulates cholesterol homeostasis and secretory pathways. However, the correlation of AnxA6 polymorphism with lipometabolism has never been studied in psoriasis. Objectives: To investigate the impact of AnxA6 polymorphism on lipid profiles and the expression of AnxA6 protein in both peripheral blood mononuclear cells (PBMCs) and lipometabolism in psoriasis. Methods: A total of 265 psoriatic patients received methotrexate (MTX) treatment for 12 weeks, after which their lipid profiles were determined by measuring total cholesterol (TC), triglycerides (TGs), lipoprotein (a) [LP(a)], high-density lipoprotein cholesterol (HDL-C), low-density lipoprotein (LDL), apolipoprotein (a)1 (ApoA1), and apolipoprotein B (ApoB). In addition, AnxA6 (*rs11960458*) was genotyped in 262 patients and the expression of AnxA6 in PBMCs was measured by Western blotting at baseline and week 8 post-MTX treatment. Results: The CC genotype carriers of *rs11960458* had a lower expression of AnxA6 and lower levels of the pro-atherogenic lipids TC, LDL, and ApoB compared to TC genotype carriers. MTX significantly downregulated the levels of the anti-atherogenic lipids HDL-C and ApoA1 and the level of AnxA6 in TC genotype carriers, as well as the level of TGs in CC genotype carriers. Conclusions: The polymorphism of AnxA6, *rs11960458*, was statistically associated with the levels of pro-atherogenic lipids and with the downregulation of MTX on the levels of anti-atherogenic lipids and TGs in psoriasis.

## 1. Introduction

Psoriasis is a chronic, recurrent, systemic inflammatory disease with multiple comorbidities. The prevalence of psoriasis in Asia ranges from 0.12% to 1.49% [1]. Common comorbidities, including cardiovascular diseases, metabolic syndrome, and high blood pressure, are associated with abnormal blood composition [2]. Evidence to date suggests that the inflammatory mediators produced by psoriatic skin lesions can enter the circulatory system and cause systemic insulin resistance and abnormal fat metabolism, eventually leading to dyslipidemia, which is a strong risk factor for atherosclerosis [3]. Dyslipidemia is not only a comorbidity of psoriasis; it is also a repercussion of psoriasis treatment with retinoids [4]. Studies have shown that psoriasis patients with arthritis (PsA) were more likely to develop dyslipidemia, as well as higher burdens of comorbid diseases, especially cardiovascular diseases [5,6].

Methotrexate (MTX) is a first-line oral, systemic therapy in treating psoriasis, especially in PsA. A clinical trial analysis showed that psoriasis patients had different efficacies to MTX treatment under different dosages, indicating that individualized drug administration is needed for MTX treatment and that an underlying genetic background might be responsible for it [7]. Previous research has found that MTX increased serum high-density lipoprotein-cholesterol (HDL-C), low-density lipoprotein (LDL), and total cholesterol (TC) levels [8]. However, other studies have reported that MTX exerted an atheroprotective effect by promoting cholesterol outflow from artery wall cells, thereby protecting patients from cardiovascular diseases [9,10]. A study in rats with adjuvant-induced arthritis reported that MTX did not affect blood lipids [11]. Thus, the effect of MTX on blood lipid profiles remains controversial, and the mechanism behind it is elusive. Considering the unique economic advantages of MTX, it is urgent to explore the mechanism underlying the clinical efficacy of MTX, thus increasing its application value.

AnnexinA6 (AnxA6) belongs to a family of calcium-dependent membrane- and phospholipid-binding proteins, and it is mainly located at the plasma membrane and endosomal compartments that serve as a scaffold protein to help recruit signaling proteins and regulate cholesterol homeostasis and adiponectin release during membrane transport [12,13]. The gene encoding AnxA6 has been reported to be one of the psoriasis susceptibility genes in the Chinese Han population [14,15,16]. Our previous study identified significant associations for the AnxA6 single nucleotide polymorphisms (SNPs) *rs11960458* and rs960709 with MTX efficacy, of which only the *rs11960458* polymorphism significantly impacted the long-term treatment outcome [17]. Another study demonstrated that elevated expression levels of protein AnxA6 could affect the intracellular distribution of cholesterol [18]. Considering the role of AnxA6 in lipometabolism, a relationship may exist between AnxA6 and the blood lipid profiles of psoriasis patients during MTX treatment. However, to date, no study has evaluated the role of the *rs11960458* polymorphism of AnxA6 in lipid changes following MTX treatment of psoriasis. Therefore, we sought to determine the impact of the *rs11960458* genotype on the expression of the AnxA6 protein, and the role of this SNP in the lipometabolism of psoriasis patients following MTX treatment.

## 2. Materials and Methods

### 2.1. Patients and Study Design

A total of 265 psoriatic patients aged ≥18 years, who received MTX treatment, and whose lipid levels were monitored, were recruited from clinics of the Dermatology Department, Fudan Huashan Hospital, between February 2015 and August 2019. Only 262 psoriatic patients were successfully genotyped as carrying the rs11960458 SNP in the *ANXA6* gene. Using the CASPAR criteria, 138 patients (52.7%) were diagnosed with psoriatic arthritis. Erythrodermic psoriasis and pustular psoriasis are not included. All patients received oral MTX therapy for 12 weeks. The medical ethics committee of Huashan Hospital approved the protocol (approval MTX201501), and all patients provided written informed consent. The diagnoses were based on typical clinical features and/or histopathological criteria. Patients who received systemic treatments (acitretin, cyclosporin, glucocorticoids) for arthritis or psoriasis at 1 month were excluded. The topical treatments had been stopped for more than one week before the beginning of the study. The therapeutic regimen followed the European guidelines on contraindications and restrictions on MTX. None of the patients used lipid-lowering drugs.

### 2.2. Treatment

The initial dose of oral MTX was 7.5–10 mg once per week, and the dose was increased by 2.5 mg every 2–4 weeks. The maximum dose was 15 mg/week, and it also depended on the patient’s clinical response, side effects, and hematological or chemical tests. If liver enzymes were elevated >2-fold and <3-fold, the dose of MTX was reduced by 2.5 mg/week and given once again 2–4 weeks later. If liver enzymes were elevated >3-fold, the use of MTX was ceased.

### 2.3. Assessment of Lipid Levels and Disease Characteristics

Two certified dermatologists graded the severity and extent of psoriasis using the Psoriasis Area Severity Index (PASI) and body surface area (BSA) scores. Lipid profiles at baseline and 12 weeks after MTX treatment, and fasting blood glucose at baseline, were measured using conventional laboratory techniques at Huashan Hospital. Sex, age, age at disease onset, smoking and alcohol intake, hypertension, diabetes, height, weight, and body mass index (BMI) were recorded.

### 2.4. DNA Extraction and Genotyping

Five milliliters of EDTA-anticoagulated whole blood were collected from all patients and stored at –80 °C. Genomic DNA was extracted from peripheral blood lymphocytes using the FlexiGene DNA Purification Kit (Qiagen, Hilden, Germany) and diluted to 20 ng/µL. All DNA samples were stored at −20 °C. The *rs11960458* SNP of AnxA6 was genotyped using a SequenomMassARRAY. The SequenomMassARRAY Assay Design 3.0 Software (Sequenom Inc., San Diego, CA, USA). was used to design the PCR parameters and detection primers. The PCR products were subsequently used as templates for locus-specific single-base extension reactions. The resulting products were desalted and transferred to a 384-element SpectroCHIP array (Sequenom Inc.). MALDI-TOF MS (Sequenom Inc.) was used for allele detection. The mass spectrograms were analyzed using MassARRAY Type software (Sequenom Inc., San Diego, CA, USA). We performed quality control of SNPs and samples at a call rate of 99.2% and analyzed the distribution of the SNPs in the HCs with the Hardy–Weinberg equilibrium (*p* > 0.0001).

### 2.5. Western Blot Assay

For SDS gel electrophoresis, samples were boiled to reduce 2×Loading buffer (120 mM Tris-HCl, 4% SDS, 20% glycerol, 2% β-mercaptoethanol, 0.2% bromophenol blue, pH 6.8). Equal protein amounts (~10 µg per lane) were separated with 10% SDS-Page using running buffer (25 mM Tris, 192 mM glycine, 0.1% SDS). After electrophoresis, gels were blotted onto PVDF membranes with the Trans-Blot Turbo transfer system (Bio-Rad, Hercules, CA, USA). Membranes were blocked for 1 h with 5% non-fat dried milk in TBST (20 mM Tris·HCl, 137 mM NaCl, 0.05% Tween-20, pH 7.6). After overnight incubation of primary antibodies ANXA6 (1:250 dilution, PA5-27462, sigma) and Tublin (1:1000 dilution, 2148, cell signaling) at 4 °C, membranes were washed and incubated with the secondary antibodies. SuperSignal West Femto chemiluminescence reagent (Pierce) was used for detection in a ChemiDoc MP system (Bio-Rad, Hercules, CA, USA), and signal intensities were quantified with Image Lab software v2.1.4.7 (National Institutes of Health, Bethesda, MD, USA).

### 2.6. Statistical Analysis

The statistical analyses were performed with the statistical software GraphPad Prism v.5 (Graph Pad Software, San Diego, CA, USA) and SPSS v.23.0 software (SPSS Inc., Chicago, IL, USA). The quantitative data were expressed as mean ± standard deviation, and qualitative data were presented as percentages. The association of genotypes with serum lipid levels was tested by covariance analysis (ANOVA). Stepwise multiple regression analysis was used to adjust sex, age, age at disease onset, disease duration, height, weight, body mass index (BMI), hypertension, diabetes, smoking and alcohol intake, and PASI score at baseline. *p*-values < 0.05 were considered statistically significant.

## 3. Results

### 3.1. The CC Genotype of AnxA6, rs11960458, Is Associated with Lower Levels of Pro-Atherogenic Lipids in Psoriatic Patients

The *rs11960458* SNP in AnxA6 was successfully genotyped in 262 of the 265 psoriatic patients. Their clinical characteristics and their lipid levels are summarized in Table 1. The levels of the pro-atherogenic lipids TC (4.56 ± 0.87 vs. 4.88 ± 0.94, *p* = 0.0215), LDL (2.79 ± 0.69 vs. 3.07 ± 0.87, *p* = 0.0161), and ApoB (0.71 ± 0.16 vs. 0.77 ± 0.18, *p* = 0.0279) in CC genotype carriers (i.e., carriers of the *rs11960458* SNP in AnxA6) were significantly lower than those in TC genotype carriers. No statistical difference was observed in other clinical characteristics and the levels of serum TGs, HDL-C, ApoA1, and Lp(a) between different genotype carriers.

As shown in Table 2, univariate analysis demonstrated that age (*p* = 0.000), age at disease onset (*p* = 0.013), gender (*p* = 0.036), arthritis (*p* = 0.048), diabetes (*p* = 0.008), and the *rs11960458* SNP (*p* = 0.012) were significantly associated with TC levels. However, only age (*p* = 0.000), gender (*p* = 0.008), and the *rs11960458* SNP (*p* = 0.015) were significantly correlated with TC levels. Furthermore, univariate analysis showed that age (*p* = 0.001), age at disease onset (*p* = 0.003), arthritis (*p* = 0.014), weight (*p* = 0.000), BMI (*p* = 0.000), hypertension (*p* = 0.002), and the *rs11960458* SNP in AnxA6 (*p* = 0.014) were significantly associated with ApoB levels. However, only age (*p* = 0.006), BMI (*p* = 0.000), and the *rs11960458* SNP (*p* = 0.039) were statistically correlated with the level of ApoB. In addition, univariate analysis revealed that diabetes (*p* = 0.002), age (*p* = 0.041), weight (*p* = 0.021), and the *rs11960458* SNP (*p* = 0.009) were significantly associated with LDL levels, but only diabetes (*p* = 0.003) and the *rs11960458* SNP (*p* = 0.021) were statistically significant.

### 3.2. The Reduction of MTX on the Levels of Anti-Atherogenic Lipids and TGs Was Associated with the rs11960458 Genotype

As shown in Table 3, MTX significantly decreased the levels of the pro-atherogenic lipids TC (*p* < 0.05), LDL (*p* < 0.05), ApoB (*p* < 0.001), and Lp(a) (*p* < 0.001) in TC and CC genotype carriers of *rs11960458* and ApoB (*p* = 0.0419) in TT genotype carriers. However, the significant downregulations of MTX treatment on the anti-atherogenic lipids HDL-C (1.19 ± 0.28 vs. 1.15 ± 0.25, *p* = 0.0068) and ApoA1 (1.06 ± 0.18 vs. 1.04 ± 0.16, *p* = 0.0434) were only observed in TC genotype carriers of *rs11960458*. Moreover, the significant reduction of MTX treatment on the level of TG was only observed in the CC genotype carriers (1.63 ± 1.04 vs. 1.41 ± 0.84, *p* = 0.0108).

### 3.3. The Level of AnxA6 Protein in PBMCs Was Significantly Increased in TC Genotype Carriers and Was Significantly Downregulated after MTX Treatment

Figure 1 shows the expression of AnxA6 protein in PBMCs from 22 psoriatic patients (4 patients with TT genotype, 9 patients with TC genotype, and 9 patients with CC genotype) before and after MTX treatment for 8 weeks. The expression of AnxA6 protein was significantly higher in TC genotype carriers of *rs11960458* than in CC genotype carriers (*p* < 0.05). MTX treatment significantly downregulated the expression of AnxA6 in TC genotypes carriers of *rs11960458* (*p* < 0.05).

## 4. Discussion

Our study demonstrated that the CC genotype carriers of *rs11960458* in AnxA6 had significantly lower levels of the pro-atherogenic lipids TC, LDL, and ApoB than TC genotype carriers. Moreover, multiple regression analysis demonstrated that the *rs11960458* SNP of AnxA6 was significantly associated with the levels of TC, LDL, and ApoB after adjusting for age, gender, age at disease onset, weight, BMI, arthritis, diabetes, and hypertension. Interestingly, the reduction of MTX treatment on the levels of the anti-atherogenic lipids HDL-C and ApoA1 was only observed in TC genotype carriers of *rs11960458*, and on the level of TG was observed in CC genotype carriers. Furthermore, the expression of AnxA6 in PBMCs was significantly higher in TC genotype carriers than in CC genotype carriers and was significantly downregulated in TC genotype carriers, after MTX treatment.

A previous study reported that AnxA6 was upregulated in the monocytes of overweight and obese patients [19]. Cholesterol efflux capacity is a functional property of HDL, reflecting the efficiency of the atheroprotective reverse cholesterol transport process in humans [20]. Several studies showed that cholesterol efflux capacity was impaired in psoriasis patients and correlated with disease severity [21,22]. Cubells et al. reported that high expression levels of AnxA6 perturbed cholesterol efflux. Therefore, the higher levels of AnxA6 protein in patients with TC genotypes of *rs11960458* may lead to impaired cholesterol efflux capacity, thereby increasing the levels of proatherogenic lipids. In addition, our results demonstrated that BMI and diabetes were positively correlated with the increased levels of ApoB and LDL, respectively, which may be due to the fact that AnxA6 stimulates endocytosis and is involved in the trafficking of LDL to the pre-lysosomal compartment [23].

Our results also demonstrated that the significant downregulation of MTX on the level of TG was only observed in CC genotype carriers of *rs11960458*. However, the expression of AnxA6 in PBMCs was not significantly changed in CC genotype carriers of *rs11960458* after MTX treatment. AnxA6 has been linked to TG storage in adipocytes in recent years. Proteomic approaches identified AnxA6 as associated with lipid droplets (LDs), located in the LD membrane of hepatocytes and adipocytes [24,25]. Another study also showed the co-localization of AnxA6 with LDs in differentiated 3T3-L1 adipocytes [26]. Stable AnxA6 overexpression increases LD numbers and size in HuH7 hepatocytes and promotes lipid accumulation in LDs. AnxA6-deficient primary hepatocytes and AnxA6 mouse liver sections display reduced LD numbers and increased serum adiponectin levels. In contrast, AnxA6 overexpression in 3T3-L1 adipocytes lowered cellular TG levels and adiponectin secretion, which means that higher levels of AnxA6 expression are associated with a higher level of lipolysis. Therefore, the decreased level of TG after MTX treatment in CC genotype carriers may be caused by the relatively low and stable expression of AnxA6, as well as the lower level of lipolysis and less lipid accumulation. Moreover, the lower expression of AnxA6 increased adiponectin secretion, which will decrease TG levels as well [27].

Furthermore, our results revealed that MTX significantly downregulated the levels of the anti-atherogenic lipids HDL-C and ApoA1 and the expression of AnxA6 in PBMCs in TC genotype carriers of *rs11960458*; however, the underlying mechanism remains unclear. Considering that cholesterol efflux capacity could be impaired by AnxA6 expression levels, the function and composition of HDL-C and ApoA1 in TT and TC genotype carriers may be dysregulated [21]. It has been reported that MTX restored cholesterol efflux capacity and, thereby, increased its protection against atherosclerosis and thrombosis [28]. MTX also regulated lipid metabolism by enhancing AMPK activation, thereby increasing the expression of adiponectin in perivascular adipose tissue [29]. Moreover, research reported that adiponectin lowers monocyte AnxA6 levels in vitro [19]. Thus, the downregulation of AnxA6 might stem from MTX, and the decreased levels of HDL-C and ApoA1 may be explained by their restored function. Overall, AnxA6 controls cholesterol and membrane transport in endocytosis and exocytosis and modulates TG accumulation and storage. 

## 5. Conclusions

In conclusion, the CC genotype of AnxA6 *rs11960458* was related to a lower level of AnxA6 protein in PBMCs and with lower levels of pro-atherogenic lipids compared to TC genotype carriers. MTX treatment significantly downregulated the expression level of AnxA6 in PBMCs and the levels of the anti-atherogenic lipids ApoA1 and HDL-C in TC genotype carriers. The mechanism of MTX treatment on the effect of blood lipids remains to be elucidated.

## Figures and Tables

**Figure 1 jcm-11-07059-f001:**
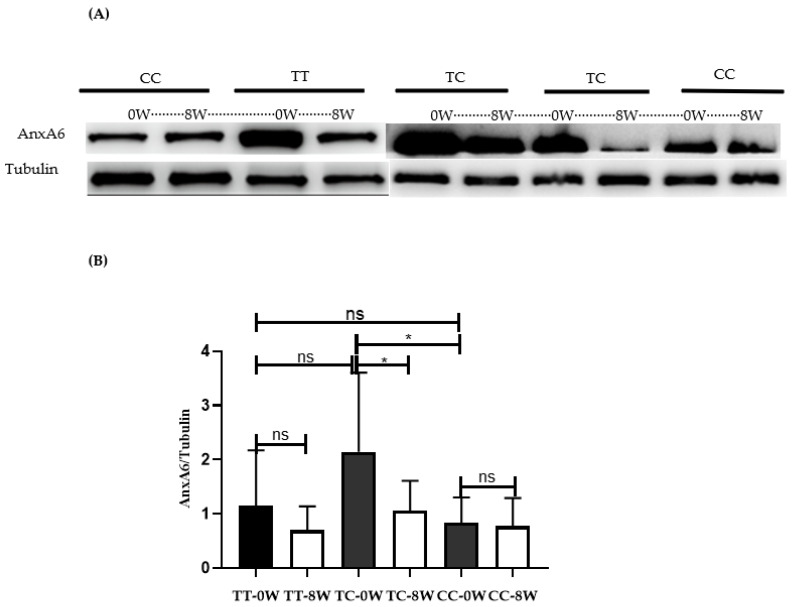
The effect of MTX and the *rs11960458* genotype on the expression of AnxA6 in PBMCs. (**A**) Western blots showing the expression of AnxA6 at baseline and at week 8, according to the *rs11960458* genotype after MTX treatment in 22 psoriatic patients (4 patients with TT genotype, 9 patients with TC genotype, and 9 patients with CC genotype); (**B**) The expression of AnxA6 in PBMCs was significantly higher in TC genotype carriers of *rs11960458* than in CC genotype carriers, and it was significantly decreased in TC genotype carriers after MTX treatment. * *p* < 0.05, one-way ANOVA (Tukey’s multiple comparisons test) and paired *t*-test were used).

**Table 1 jcm-11-07059-t001:** Clinical characteristic and the lipid levels according to the *rs11960458* genotype of AnxA6 in 262 psoriatic patients.

	TT (*n* = 40)	TC (*n* = 127)	CC (*n* = 95)	*p*-Value
Age, mean (SD), year	50.8 ± 16.62	47.94 ± 14.60	46.35 ± 14.96	0.2903
Age at disease onset, mean (SD), year	36.47 ± 16.65	34.34 ± 15.99	34.27 ± 15.12	0.7286
Disease duration, mean (SD), year	14.33 ± 11.17	13.59 ± 10.69	12.2 ± 10.28	0.4800
Weight, mean (SD), kg	69.86 ± 13.48	68.41 ± 11.83	69.03 ± 12.77	0.8077
BMI, kg/m^2^	24.91 ± 3.67	24.26 ± 3.39	24.29 ± 3.71	0.5866
PASI score at baseline	12.21 ± 6.05	13.65 ± 7.99	15.4 ± 7.50	0.0580
The mean PASI improvement at 12 week	59.7 ± 32.05	60.34 ± 31.88	68.84 ± 31.7	0.1082
PASI 50 response at 12 w	25 (62.50)	87 (68.50)	73 (76.84)	0.1904
PASI 75 response at 12 week	19 (47.50)	52 (40.94)	51 (53.68)	0.1685
PASI 90 response at 12 week	7 (17.50)	24 (18.90)	26 (27.37)	0.2474
Male, *n* (%)	31(77.50)	91 (71.65)	63 (66.320	0.4013
Arthritis, *n* (%)	24 (60.00)	66 (51.97)	48 (50.52)	0.5879
Smoking, *n* (%)	14 (35.00)	38 (29.92)	29 (30.52)	0.1795
Hypertension, *n* (%)	21 (52.50)	53 (41.73)	32 (33.68)	0.1163
Diabetes, *n* (%)	10 (25.00)	29 (22.83)	14 (14.74)	0.2376
TC, mmol/L	4.92 ± 0.97	4.88 ± 0.94	4.56 ± 0.87 *	**0.0215**
TG, mmol/L	1.73 ± 0.98	1.56 ± 0.88	1.63 ± 1.04	0.6122
HDL-C, mmol/L	1.20 ± 0.26	1.19 ± 0.28	1.16 ± 0.31	0.5863
LDL-C, mmol/L	3.11 ± 0.76	3.07 ± 0.87	2.79 ± 0.69 *	**0.0161**
ApoA1, g/L	1.06 ± 0.18	1.06 ± 0.18	1.03 ± 0.18	0.5198
ApoB, g/L	0.78 ± 0.15	0.77 ± 0.18	0.71 ± 0.16 *	**0.0279**
Lp(a), mg/L	156.6 ± 176.5	165.4 ± 196.9	138.1 ± 133.3	0.5097

Abbreviations: BMI, body mass index (kg/m^2^); BSA, body surface area; PASI, Psoriasis Area Severity Index; TC, total cholesterol; TG, triglyceride; HDL-C, high-density lipoprotein-cholesterol; LDL, low-density lipoprotein; ApoA1, apolipoprotein A1; ApoB, apolipoprotein B; Lp(a), lipoprotein (a). Data are presented as the number (percentage) of patients unless otherwise indicated. One-way ANOVA or Chi-square was used when appropriate. *p* < 0.05 was considered statistically significant. Significant *p*-values are shown in bold. * represents a significant difference between TC- and CC-genotype carriers.

**Table 2 jcm-11-07059-t002:** Univariate and multivariate analyses of clinical factors and AnxA6 polymorphism associated with the levels of pro-atherogenic lipids in 262 psoriatic patients.

		Univariate Analysis		Multivariate Analysis	
	Predictors	OR (95%CI)	*p*-Value	OR (95%CI)	*p*-Value
TC	*rs11960458*	−0.209 (−0.372~−0.046)	**0.012**	−0.198 (−0.358~−0.039)	**0.015**
	gender	−0.264 (−0.511~−0.017)	**0.036**	−0.324 (−0.563~−0.0830)	**0.008**
	age	0.0137 (0.006~0.021)	**0.000**	0.0136 (0.006~0.021)	**0.000**
	arthritis	0.228 (0.002~0.454)	**0.048**		
	onset	0.009 (0.002~0.0160	**0.013**		
	diabetes	0.381 (0.103~0.660)	**0.008**		
ApoB	*rs11960458*	−0.037 (−0.067~−0.007)	**0.014**	−0.031 (−0.061~−0.002)	**0.039**
	age	0.002 (0.001~0.004)	**0.001**	0.002 (0.001~0.003)	**0.006**
	BMI	0.011 (0.005~0.016)	**0.000**	0.010 (0.005~0.015)	**0.000**
	arthritis	0.052 (0.011~0.093)	**0.014**		
	weight	0.003 (0.001~0.004)	**0.000**		
	onset	0.002 (0.001~0.003)	**0.003**		
	hypertension	0.067 (0.026~0.109)	**0.002**		
LDL	*rs11960458*	−0.187 (−0.328~0.047)	**0.009**	−0.169 (−0.312~−0.026)	**0.021**
	diabetes	0.380 (0.142~0.619)	**0.002**	0.367 (0.124~0.610)	**0.003**
	age	0.007 (0.000~0.013)	**0.041**		
	weight	0.009 (0.001~0.016)	**0.021**		

Abbreviations: TC, total cholesterol; LDL: low-density lipoprotein; ApoA1, apolipoprotein A1; ApoB, apolipoprotein B; Lp(a), lipoprotein (a). Multiple regression analysis was performed after adjustment for age, age at disease onset, disease duration, weight, BMI, hypertension, diabetes, arthritis, and the genotype of *rs11960458* in AnxA6. Only the significant variables are shown. *p* < 0.05 is considered statistically significant (shown in bold).

**Table 3 jcm-11-07059-t003:** The effect of 12-week MTX treatment on lipid levels according to the genotype of AnxA6 *rs11960458* in 262 psoriatic patients.

	TT (*n* = 40)			TC (*n* = 127)			CC (*n* = 95)		
	0 week	12 week	*p*-Value	0 week	12 week	*p*-Value	0 week	12 week	*p*-Value
TC, mmol/L	4.92 ± 0.97	4.69 ± 1.02	0.0816	4.88 ± 0.94	4.65 ± 0.94	**<0.001**	4.56 ± 0.87	4.40 ± 0.80	**0.0111**
TG, mmol/L	1.73 ± 0.98	1.59 ± 0.91	0.1344	1.56 ± 0.88	1.64 ± 1.03	0.2784	1.63 ± 1.04	1.41 ± 0.84	**0.0108**
HDL-C, mmol/L	1.20 ± 0.26	1.16 ± 0.26	0.4109	1.19 ± 0.28	1.15 ± 0.25	**0.0068**	1.16 ± 0.31	1.17 ± 0.32	0.4519
LDL, mmol/L	3.11 ± 0.76	2.97 ± 0.82	0.1501	3.07 ± 0.87	2.94 ± 0.86	**0.0028**	2.79 ± 0.69	2.68 ± 0.62	**0.0353**
ApoA1, g/L	1.06 ± 0.18	1.04 ± 0.15	0.4569	1.06 ± 0.18	1.04 ± 0.16	**0.0434**	1.03 ± 0.18	1.02 ± 0.18	0.2467
ApoB, g/L	0.78 ± 0.15	0.73 ± 0.16	**0.0419**	0.77 ± 0.18	0.72 ± 0.16	**<0.001**	0.71 ± 0.16	0.67 ± 0.15	**0.0001**
Lp(a), mg/L	156.6 ± 176.5	133.5 ± 137.9	0.0503	165.4 ± 196.9	149.2 ± 176.9	**0.0069**	138.1 ± 133.3	120.5 ± 130.2	**0.0005**

Abbreviation: TC, total cholesterol; TG, triglyceride; HDL-C, high-density lipoprotein-cholesterol; LDL, low-density lipoprotein; ApoA1, apolipoprotein A1; ApoB, apolipoprotein B; Lp(a), lipoprotein (a). Paired *t*-test was used. *p* < 0.05 is considered to be statistically significant. Significant *p* values are shown in bold.

## Data Availability

The dataset used and analyzed in this present study is available from the corresponding author.

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
