# Peer review of "Annexin A6 Polymorphism Is Associated with Pro-atherogenic Lipid Profiles and with the Downregulation of Methotrexate on Anti-Atherogenic Lipid Profiles in Psoriasis"

_jcm, 2022, doi:10.3390/jcm11237059_

Round 1
Reviewer 1 Report
I think this paper has many and much important problem to publish. Especially since, there is doubt in the statistical analysis, if its accuracy is not ensured, there can be no discussion.
At line 37-38
The cited paper and this sentence seemed completely unrelated.
At 137
No comparison should be made between groups without significant differences in the analysis of variance. This paper need to be checked by an expert of statistical analysis.
I'm sorry I don't understand the results of Table 2 at all and whoud like to know more. According to this paper, the polymorphism of rs11960458 in AnxA6 are 262 out of 265 cases. Is this Table2 result based on a comparison between 262 cases and 3 cases? Or is it a comparison between CC genotype and the others? Please let me see the raw data.
Line 170-1
The results of this text are not shown. It shoul be submitted as a supplement table. Or the overall data should also be added to Table3.
Line 186-7
Please tell me which mulple comparison method you used. I would like to see the raw data of this results of this analysis.
Line 134-6,170-1
I would like to knowe what test was used to derive this result, it is not stated in method part.
Author Response
I think this paper has many and much important problem to publish. Especially since, there is doubt in the statistical analysis, if its accuracy is not ensured, there can be no discussion.
Answer: Thank you for your constructive suggestion. We found that a wrong version was uploaded when we changed the format of this text. I apologize for our mistake.
At line 37-38
The cited paper and this sentence seemed completely unrelated.
Answer; I am very sorry for our mistake. We had changed this sentence according to your suggestion.
Psoriasis is a chronic, recurrent, systemic inflammatory disease with multiple comorbidities. The prevalence of psoriasis in Asia ranged from 0.12 to 1.49% [1].
At 137
No comparison should be made between groups without significant differences in the analysis of variance. This paper need to be checked by an expert of statistical analysis.
Answer: This paper has been seen by an expert (professor Jianfeng Luo) of statistical analysis and have we adapted the statistical analysis according to the suggestion of previous reviewer.
The new results are shown below
The levels of pro-atherogenic lipids TC (4.56±0.87 vs 4.88±0.94, p= .0215), LDL (2.79±0.69 vs 3.07±0.87, p= .0161), and ApoB (0.71±0.16 vs 0.77±0.18, p=0.0279) in CC genotype carriers of rs11960458 in AnxA6 were significantly lower than those with TC genotype carriers.
New Table1
|
TT (n = 40) |
TC (n = 127) |
CC (n = 95) |
p-value |
|
|
Age, mean (SD), y |
50.8 ± 16.62 |
47.94 ± 14.60 |
46.35 ± 14.96 |
0.2903 |
|
Age at disease onset, mean (SD), y |
36.47 ± 16.65 |
34.34 ± 15.99 |
34.27 ± 15.12 |
0.7286 |
|
Disease duration, mean (SD), y |
14.33 ± 11.17 |
13.59 ± 10.69 |
12.2 ± 10.28 |
0.4800 |
|
Weight, mean (SD), kg |
69.86 ± 13.48 |
68.41 ± 11.83 |
69.03 ± 12.77 |
0.8077 |
|
BMI, kg/m2 |
24.91 ± 3.67 |
24.26 ± 3.39 |
24.29 ± 3.71 |
0.5866 |
|
PASI score at baseline |
12.21 ± 6.05 |
13.65 ± 7.99 |
15.4 ± 7.50 |
0.0580 |
|
The mean PASI improvement at 12 w |
59.7 ± 32.05 |
60.34 ± 31.88 |
68.84 ± 31.7 |
0.1082 |
|
PASI 50 response at 12 w |
25 (62.50) |
87 (68.50) |
73 (76.84) |
0.1904 |
|
PASI 75 response at 12 w |
19 (47.50) |
52 (40.94) |
51 (53.68) |
0.1685 |
|
PASI 90 response at 12 w |
7 (17.50) |
24 (18.90) |
26 (27.37) |
0.2474 |
|
Male, n (%) |
31(77.50) |
91 (71.65) |
63 (66.320 |
0.4013 |
|
Arthritis, n (%) |
24 (60.00) |
66 (51.97) |
48 (50.52) |
0.5879 |
|
Smoking, n (%) |
14 (35.00) |
38 (29.92) |
29 (30.52) |
0.1795 |
|
Hypertension, n (%) |
21 (52.50) |
53 (41.73) |
32 (33.68) |
0.1163 |
|
Diabetes, n (%) |
10 (25.00) |
29 (22.83) |
14 (14.74) |
0.2376 |
|
TC, mmol/L |
4.92 ± 0.97 |
4.88 ± 0.94 |
4.56 ± 0.87* |
0.0215 |
|
TG, mmol/L |
1.73 ± 0.98 |
1.56 ± 0.88 |
1.63 ± 1.04 |
0.6122 |
|
HDL-C, mmol/L |
1.20 ± 0.26 |
1.19 ± 0.28 |
1.16 ± 0.31 |
0.5863 |
|
LDL-C, mmol/L |
3.11 ± 0.76 |
3.07 ± 0.87 |
2.79 ± 0.69* |
0.0161 |
|
ApoA1, g/L |
1.06 ± 0.18 |
1.06 ± 0.18 |
1.03 ± 0.18 |
0.5198 |
|
ApoB, g/L |
0.78 ± 0.15 |
0.77 ± 0.18 |
0.71 ± 0.16* |
0.0279 |
|
Lp(a), mg/L |
156.6 ± 176.5 |
165.4 ± 196.9 |
138.1 ± 133.3 |
0.5097 |
Previous Table1
|
rs11960458 |
|||
|
TT+TC (n=167) |
CC (n=95) |
p-value |
|
|
Age, mean (SD), y |
48.63±15.11 |
46.35±14.96 |
.2394 |
|
Age at disease onset, mean (SD), y |
34.85±16.13 |
34.27±15.12 |
.7768 |
|
Disease duration, mean (SD), y |
13.77±10.77 |
12.2±10.28 |
.2501 |
|
Weight, mean (SD), kg |
68.76±12.22 |
69.03±12.77 |
.8695 |
|
BMI, kg/m2 |
24.42±3.46 |
24.29±3.71 |
.7818 |
|
PASI score at baseline |
13.31±7.58 |
15.4±7.50 |
.0319 |
|
The mean PASI improvement at 12W |
60.19±31.83 |
68.84±31.7 |
.035 |
|
Male, n (%) |
122 (73.05) |
63 (66.32) |
.2619 |
|
Arthritis, n (%) |
90 (53.89) |
48 (50.53) |
.6093 |
|
Hypertension, n (%) |
74 (44.31) |
32 (33.68) |
.116 |
|
Diabetes, n (%) |
39 (23.35) |
14 (14.74) |
.1106 |
|
Smoking, n (%) |
52 (31.14) |
29 (30.53) |
>.99 |
|
TC, mmol/L |
4.89±0.95 |
4.56±0.87 |
.0057 |
|
TG, mmol/L |
1.60±0.90 |
1.63±1.04 |
.8433 |
|
HDL-C, mmol/L |
1.19±0.28 |
1.16±0.31 |
.3011 |
|
LDL-C, mmol/L |
3.08±0.84 |
2.79±0.69 |
.0042 |
|
ApoA1, g/L |
1.06±0.18 |
1.03±0.18 |
.2574 |
|
ApoB, g/L |
0.77±0.17 |
0.71±0.16 |
.0078 |
|
Lp(a), mg/L |
163.3±191.8 |
138.1±133.3 |
.2593 |
I'm sorry I don't understand the results of Table 2 at all and whoud like to know more. According to this paper, the polymorphism of rs11960458 in AnxA6 are 262 out of 265 cases. Is this Table2 result based on a comparison between 262 cases and 3 cases? Or is it a comparison between CC genotype and the others? Please let me see the raw data.
Answer: Thank you for your questions. I found that I had upload a false version when we changed the format of this text. We analyzed the association of the genotypes (TT, TC, CC) of rs11960458 in AnxA6 with the pro-atherogenic lipids.
As shown in Table.2, univariate analysis demonstrated that age (p= .000), age at disease onset (p= .013), gender (p= .036), arthritis (p= .048), diabetes (p= .008), and the polymorphism of rs11960458 (p= .012) were statistically associated with the level of TC. But only age (p= .000), gender (p=0.008) and the polymorphism of rs11960458 (p= .015) were statistically correlated with the level of TC. Furthermore, univariate analysis showed that age (p= .001), age at disease onset (p= .003), arthritis (p= .014), weight (p= .000), BMI (p= .000), hypertension (p= .002), and the polymorphism of rs11960458 in AnxA6 (p= .014) were statistically associated with the level of ApoB. But only age (p= .006), BMI (p= .000), and the polymorphism of rs11960458 (p=.039) were statistically correlated with the level of ApoB. In addition, univariate analysis found that diabetes (p= .002), age (p= .041), weight (p= .021), and the polymorphism of rs11960458 (p= .009) were statistically associated with the level of LDL. But only diabetes (p= .003) and the polymorphism of rs11960458 (p= .021) were statistically significant.
Line 170-1
The results of this text are not shown. It shoul be submitted as a supplement table. Or the overall data should also be added to Table3.
Answer: Thank you for your questions. I found that I had upload a false version when we changed the format of this text.
As can been seen from Table 3, MTX significantly decreased the levels of pro-athero genic lipids TC (p< .05), LDL (p< .05), ApoB (p< .001), and Lp(a) (p< .001) in TC and CC genotype carriers of rs11960458 and ApoB (p= .0419) in TT genotype carriers. However, the significant downregulations of MTX treatment on the anti-atherogenic lipids HDL-C (1.19±0.28 vs 1.15±0.25, p= .0068) and ApoA1 (1.06±0.18 vs 1.04±0.16, p= .0434) were only observed in TC genotype carriers of rs11960458. Moreover, the significant reduction of MTX treatment on the level of TG was only observed in the CC genotype carriers (1.63±1.04 vs 1.41±0.84, p= .0108).
Line 186-7
Please tell me which mulple comparison method you used. I would like to see the raw data of this results of this analysis.
Answer: Turkey’s multiple comparions test was used when we compared the difference of AnxA6 protein expression between TT, TC, and CC genotypes at baseline. Paired t-test was used when we compared the difference of AnxA6 protein expression before and after MTX treatment.
The raw data of this results is the following
|
TT-0W |
TT-8W |
TC-0W |
TC-8W |
CC-0W |
CC-8W |
|
0.70557 |
0.533364 |
0.928333 |
0.49652 |
0.717372 |
0.500265 |
|
0.488268 |
0.472281 |
4.045427 |
2.141596 |
1.048333 |
0.804217 |
|
0.762538 |
0.432085 |
0.790908 |
0.555223 |
1.088471 |
0.800864 |
|
2.667863 |
1.353475 |
0.83151 |
1.003774 |
1.817883 |
1.929801 |
|
3.928034 |
1.227791 |
0.932497 |
0.96778 |
||
|
0.865647 |
1.294367 |
0.666905 |
0.176575 |
||
|
1.741114 |
0.412276 |
0.4380296 |
0.8439921 |
||
|
4.019598 |
0.974672 |
0.6091067 |
0.7351749 |
||
|
2.112767 |
1.433322 |
0.2284026 |
0.1861261 |
Line 134-6,170-1
I would like to know what test was used to derive this result, it is not stated in method part.
Answer: I am sorry for our mistake that we uploaded a wrong version when we changed the format of this text.
We had added the statistical method in the following of Tables.
As can been seen from Table 3, MTX significantly decreased the levels of pro-athero genic lipids TC (p< .05), LDL (p< .05), ApoB (p< .001), and Lp(a) (p< .001) in TC and CC genotype carriers of rs11960458 and ApoB (p= .0419) in TT genotype carriers. However, the significant downregulations of MTX treatment on the anti-atherogenic lipids HDL-C (1.19±0.28 vs 1.15±0.25, p= .0068) and ApoA1 (1.06±0.18 vs 1.04±0.16, p= .0434) were only observed in TC genotype carriers of rs11960458. Moreover, the significant reduction of MTX treatment on the level of TG was only observed in the CC genotype carriers (1.63±1.04 vs 1.41±0.84, p= .0108).
Reviewer 2 Report
Overall comments
As a clinical dermatologist with a research interest in genomics and metabolics I found this paper to make for good reading.
I believe the clinical characteristics can be embellished by including other metabolic paraments (ex waist circumference), indices (atherogenic index of plasma, possibly can be calculated from available data) and investigations (Carotid artery intima thickness). The latter could possibly be grounds for a further study.
Minor
Line 42: Dyslipidaemia is not only a comorbidity of psoriasis but also a repercussion of ….
Line 47: I would specify that MTX is a first line oral, systemic therapy (as one would assume first line treatments to be topicals and NB-UVB).
Line 78: rewords “detected the levels”
Line 82: Erythrodermic psoriasis
Line 184: Figure one shows
Line 236: consider rewording
Author Response
Overall comments
As a clinical dermatologist with a research interest in genomics and metabolics I found this paper to make for good reading.
I believe the clinical characteristics can be embellished by including other metabolic paraments (ex waist circumference), indices (atherogenic index of plasma, possibly can be calculated from available data) and investigations (Carotid artery intima thickness). The latter could possibly be grounds for a further study.
Answer: Thank you for your constructive suggestion. I agree with you that the elements including waist circumference and carotid artery intima thickness are of great importance. Actually, we are improving the construction of psoriasis specific disease database system. Since June 2021, we have collected the data of the height, weight, BMI, systolic blood pressure and diastolic blood pressure from all patients. In addition, since 2022, Carotid artery B-ultrasound has been detected in all new patients. However, presently, the patient history information collected is not complete. Therefore, we will conduct more in-depth research in the future.
Minor
Line 42: Dyslipidaemia is not only a comorbidity of psoriasis but also a repercussion of ….
Anwer: Thank you for your good suggestion. We had revised this sentence according to your suggestion.
Dyslipidemia is not the only comorbidity of psoriasis but also a repercussion of the treatment of psoriasis with retinoids
Line 47: I would specify that MTX is a first line oral, systemic therapy (as one would assume first line treatments to be topicals and NB-UVB).
Answer: We had changed this sentence according to your suggestion.
Methotrexate (MTX) is a first line oral, systemic therapy in treating psoriasis, especially in PsA.
Line 78: rewords “detected the levels”
Answer: We had reworded this sentence according to your suggestion.
A total of 265 psoriatic patients aged ≥18 years, who received MTX treatment and whose levels of lipid profiles were monitored, were recruited from clinics of the Dermatology Department, Fudan Huashan Hospital, between February 2015 and August 2019, but only 262 psoriatic patients were successfully genotyped in AnxA6 gene rs11960458.
Line 82: Erythrodermic psoriasis
Answer: We had changed this sentence according to your suggestion.
Erythrodermic psoriasis and pustular psoriasis are not included.
Line 184: Figure one shows
Answer: We had changed this sentence according to your suggestion.
Figure one shows the expression of AnxA6 protein in PBMCs from 22 psoriatic patients (4 patients with TT genotype, 9 patients with TC genotype, 9 patients with CC genotype) before and after MTX treatment for 8 weeks.
Line 236: consider rewording
Answer: We have reworded this sentence.
In addition, our results demonstrated that the significant downregulation of MTX on the level of TG was only observed in CC genotype carriers of rs11960458.
Reviewer 3 Report
The study is very interesting and can have a great clinical impact. The great advantage of the study is the large study group, however no control group was recruited. In the clinical characteristic of the study group the presence of cardiovascular diseases should be involved. The authors have written that studies have shown that psoriasis patients with PsA were more likely to develop dyslipidemia and cardiovascular diseases. It would be improving the quality of work if the levels of pro-atherogenic lipids were analyzed additionally in the PsA group of patients treated with MTX. The presentation of results only in table is not appropriate for the readers. In the Table 1 legend the use of “*” is not explained. In the Table 2 the data about the analysis LDL vs rs11960458 are not complete (“-0.187 (-0.328~”).
Author Response
The study is very interesting and can have a great clinical impact. The great advantage of the study is the large study group, however no control group was recruited. In the clinical characteristic of the study group the presence of cardiovascular diseases should be involved. The authors have written that studies have shown that psoriasis patients with PsA were more likely to develop dyslipidemia and cardiovascular diseases. It would be improving the quality of work if the levels of pro-atherogenic lipids were analyzed additionally in the PsA group of patients treated with MTX. The presentation of results only in table is not appropriate for the readers. In the Table 1 legend the use of “*” is not explained. In the Table 2 the data about the analysis LDL vs rs11960458 are not complete (“-0.187 (-0.328~”).
Answer:Thank you for your good suggestion. However, only 5 patients (1.9%) complained of coronary artery disease, no myocardial infarction and stroke were reported in our cohort. Therefore, we only observed the incidence of hypertension in the clinical characteristics. We had reported the effect of MTX on the levels of pro-atherogenic lipids in another published paper (Wang B, Deng H, Hu Y, Han L, Huang Q, Fang X, Yang K, Wu S, Zheng Z, Yawalkar N, Zhang Z, Yan K, The difference of lipid profiles between psoriasis with arthritis and psoriasis without arthritis and sex-specific downregulation of methotrexate on the apolipoprotein B/apolipoprotein A-1 ratio. Arthritis research & therapy 2022;24: 17.) Our results demonstrated that the downregulations of MTX on the level of LDL and ApoA1 were only observed in PsO patients, and the downregulation of MTX on the level of TG was only observed in PsA patients.
We are sorry for our mistake that “*” is not explained in table 1, and the data about the analysis LDL vs rs11960458 are not complete (“-0.187 (-0.328~”) in Table 2.
We have added “* represents the statistical difference between TC and CC genotype carriers” in the following of Table 1.
We have completed the data” -0.187 (-0.328~0.047) “ in Table 2.